# Association between maternal adversity, DNA methylation, and cardiovascular health of offspring: a longitudinal analysis of the ALSPAC cohort study

Natalie K Hyde [ID],[1] James G Dowty,[2] Anna Scovelle,[3] Gregory Armstrong [ID],[4] Georgina Sutherland,[4] Lisa Olive,[1,5] Kate Lycett,[5,6] Adrienne O'Neil[1]

For numbered affiliations see end of article.

**Correspondence to**
Dr Natalie K Hyde;
natalie.hyde@deakin.edu.au

## ABSTRACT

**Objectives** Maternal adversity during pregnancy has been shown to be associated with some health outcomes in the offspring. This study investigated the association of maternal adversity during pregnancy and DNA methylation with offspring cardiovascular (CV) health.

**Design** Longitudinal observational cohort study

**Setting** All pregnant residents in county Avon (~0.9 million), UK, were eligible to participate if their estimated delivery date was between 1 April 1991 and 31 December 1992.

**Participants** Mother–offspring pairs enrolled in the Avon Longitudinal Study of Parents and Children cohort at seven (n=7431) and 17 years of age (n=3143).

**Primary and secondary outcome measures** Offspring CV health primary measures were heart rate (HR), blood pressure (BP) and secondary measures were pulse-wave velocity and carotid intima–media thickness.

**Results** Overall, there was no association between maternal adversity scores (number or perceived impact) and primary CV measures (Perceived impact; HR: 0.999-fold change 95% CI 0.998 to 1.001; systolic BP (SBP): 1.000-fold change 95% CI 0.999 to 1.001; diastolic BP: 1.000-fold change 95% CI 0.999 to 1.002). Some small offspring sex effects were observed and there was also a small association between methylation of some CpG sites and offspring BP measures.

**Conclusions** We found little evidence to support the overall association of maternal adversity during pregnancy and DNA methylation with offspring CV measures. Offspring sex-specific and age-specific associations require further investigation.

## INTRODUCTION

Seminal work conducted by Barker *et al* in the early 20th century noted geographical differences in infant mortality rates, whereby regions of England with the highest infant mortality rates also had the highest rate of coronary heart disease (CHD) mortality.[1 2] From this, it was concluded '…adverse environmental influences in utero and during infancy, associated with poor living standards, directly increased susceptibility to the disease

### Strengths and limitations of this study

► A strength of this study is longitudinal collection of phenotypic data in both women and their child; detailed cardiovascular measures in the offspring have been collected at multiple time points.

► A limitation is attrition bias, with those of a higher socioeconomic status being more likely to remain in the study over time.

► In addition it is possible that some life stressors during pregnancy may have not have been captured given the list of potential stressors was not exhaustive.

► It is plausible that any effects during pregnancy may have been diluted by the inclusion of data about maternal stressors that was collected in the early postpartum phase (8 weeks post partum).

(CHD).' Further extending this work, the Developmental Origins of Health and Disease hypothesis[3] proposes that the risk of chronic diseases originate not only from an individual's genome but also by its interactions with biological insults in utero and early life.

To date, much of the work in this area has focused on the impact of maternal nutrition during pregnancy, with comparatively fewer data on social adversity and trauma. However, there are some data to suggest that the ways in which women experience social adversity during pregnancy may induce similar changes to disease trajectory in the offspring as maternal malnourishment.[4]

The time in utero represents a critical period of development, which may be particularly vulnerable to maternal stress. During this time, it is plausible the fetus is directly susceptible to the biological effects of maternal stress owing to its reliance on the maternal blood supply via the placenta. Epigenetics are mitotically heritable changes to gene expression that do not involve changes to the underlying

genetic sequence. These changes in gene expression may provide some clues about the mechanisms through which maternal adversity embeds itself into an individual and her offspring. A recent review of maternal prenatal stress and infant DNA methylation identified several candidate genes implicated in the maternal central stress response that may be critical in driving phenotype changes for offspring.[4] However, recent evidence also suggests that perhaps the placenta may buffer the effects of the maternal stress response.[5]

The extent of cumulative damage to biological systems that occurs with increasing number, duration or severity of exposures, particularly with age, is likely to be a critical consideration in understanding associations between maternal adversity and the cardiovascular (CV) health of a child. This includes distinguishing the response (eg, perceived stress) from the stimulus or stressor (eg, the adversity) itself. It is also notable that there appears to be a sexually dimorphic response with regards to several developmental exposures and CV conditions.[6] These issues, along with other key gaps in the evidence base that exist in psycho-cardiology have been outlined in the position paper by the American Heart Association (AHA).[7] Specifically, these gaps relate to, (1) an absence of truly prospective studies that commenced in the prenatal period with capacity to explicate this relationship and (2) a lack of studies that identify the biological mechanisms linking adversity to CV disease (CVD).

This study therefore seeks to, (1) investigate the respective and cumulative impact of women's exposure(s) to adversity during the perinatal period, and cord blood DNA methylation on CV health of her offspring, (2) establish whether associations are sex or age-dependent and (3) determine whether DNA methylation at birth is associated with CV outcomes. We hypothesise that greater maternal adversity will be associated with poorer CV health of offspring and DNA methylation will be associated with offspring CV measures.

## METHODS
### Study design and participants
This study used longitudinal data from the Avon Longitudinal Study of Parents and Children (ALSPAC formerly 'Children of the 90s' study). ALSPAC is a prospective birth cohort study conducted in the UK. The full study protocol is available elsewhere with participation rates and reason for not participation.[8 9] Briefly, all pregnant women residing in county Avon (~0.9 million) were eligible to participate if their estimated delivery date was between 1 April 1991 and 31 December 1992 inclusive. Recruitment occurred via maternity health services and mass media campaigns. After their initial expression of interest and assessment of eligibility by ALSPAC staff, women were sent the baseline questionnaire ~~1 week later. The women of 14541 pregnancies (71.8% of all pregnancies in the area at that time) were recruited antenatally during 1990–1992. They completed a series of postal questionnaires throughout their pregnancy and there were several clinical assessments postbirth. CV health data were collected when the children were aged 7 and 17 years.

There were 13617 mother–offspring pairs from singleton live births who survived to ≥1 year of age; only singleton pregnancies and those women with term deliveries were included in the analyses. When the oldest children were approximately 7 years of age, an attempt was made to bolster the initial sample with eligible cases who had failed to join the study originally, resulting in an additional 913 children being enrolled. The total sample size for analyses using any data collected after the age of 7 is therefore 15454 pregnancies, resulting in 15589 fetuses. Of these 14901 were alive at 1 year of age. The number of children with CV measures at the subsequent 7-year and 17 year time points were 7431 and 5215 (figure 1), respectively, and were included in analyses if they had complete information for the relevant analyses.

### Measures
#### Exposure variable
Data were those provided by women at (1) 0–18 weeks gestation, and (2) between 19 weeks gestation and 8 weeks post partum. Women retrospectively self-reported social adversities and rated its impact for the respective period. Adverse life events were assessed using a 41-item self-report questionnaire based on a Life Events Inventory,[10] using the average score at the two timepoints. The internal reliability of the inventory, as indicated by the coefficient, is 0.68. Each item was rated in one of five categories: 'Yes, affected me a lot,' 'Yes, affected me moderately,' 'Yes, affected me mildly,' 'Yes, but did not affect me' and 'No, did not happen' and was rated from 0 to 4, with higher scores indicating greater perceived stress. Two scores were calculated as follows: (1) the number of stressful life and (2) the perceived impact of the events.

#### Outcome variables
The primary outcomes were blood pressure (BP) and heart rate (HR) at 7 and 17 years of age. Duplicate measures of resting HR, systolic BP (SBP) and diastolic BP (DBP) were taken using a Dinamap 9301 Vital Signs Monitor while participants were seated, using the average of the two readings.

Secondary outcomes were pulse-wave velocity (PWV) and carotid intima media thickness (cIMT) at 17 years of age, using the mean of three measures and the mean of three end-diastolic measurements of both the left and right side, respectively. A Vicorder device (Skidmore Medical, UK) was used to measure PWV and a Zonare Z.OneUltra system that had a a L10-5 linear transducer (Zonare Medical Systems, CA, US) was used to determine cIMT. Detailed protocols have been described elsewhere.[11]

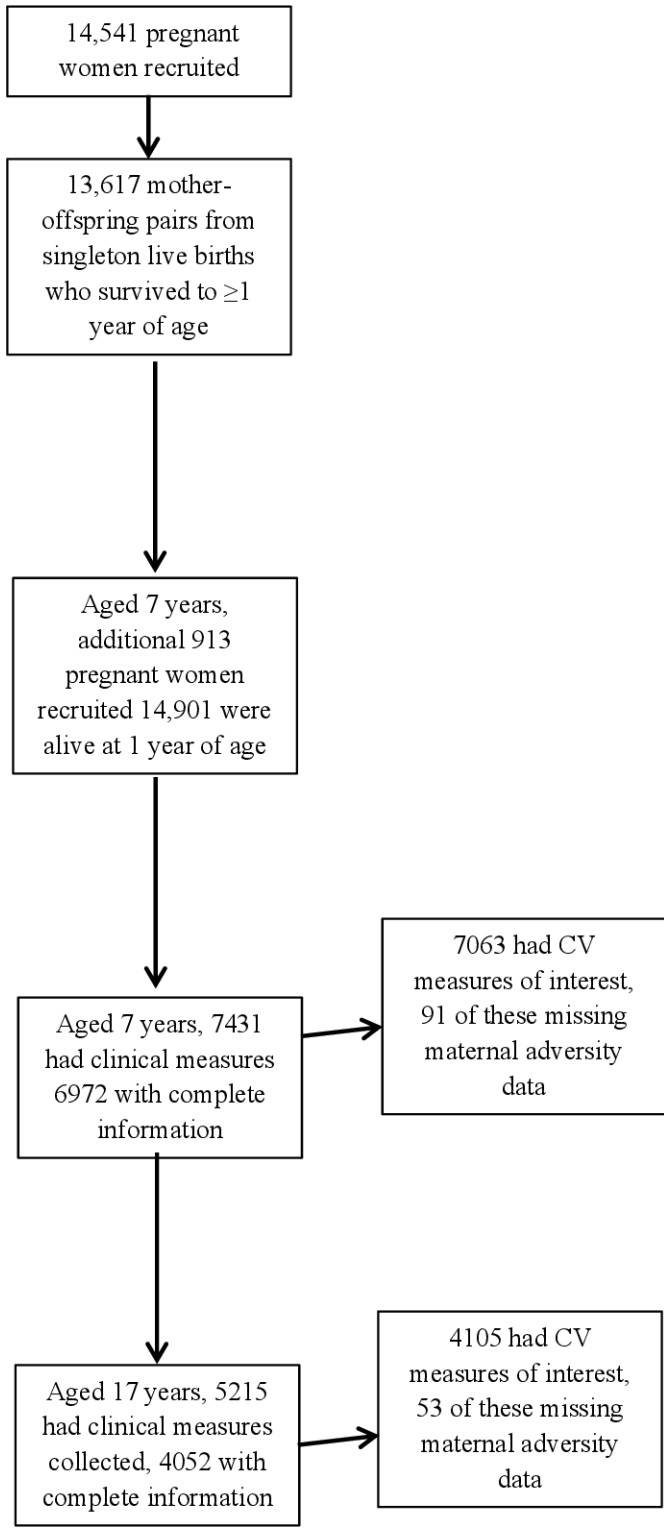

**Figure 1** Participation flow chart.

Text within the flowchart:

14,541 pregnant women recruited

13,617 mother-offspring pairs from singleton live births who survived to ≥1 year of age

Aged 7 years, additional 913 pregnant women recruited 14,901 were alive at 1 year of age

Aged 7 years, 7431 had clinical measures 6972 with complete information

7063 had CV measures of interest, 91 of these missing maternal adversity data

Aged 17 years, 5215 had clinical measures collected, 4052 with complete information

4105 had CV measures of interest, 53 of these missing maternal adversity data

### DNA methylation data

Embedded within the ALSPAC study is a human epigenetic resource; the Accessible Resource for Integrated Epigenomic Studies (ARIES).[12] Of the 1018 mother–offspring pairs in the ARIES project, 916 offspring had cord blood methylation data, which passed quality control.[13] Venous cord blood at birth was used to assess epigenome-wide methylation levels using the Illumina Infinium Human-Methylation450 BeadChip.

Raw intensity signals were processed and M-values were calculated using the minfi package.[14] Probes and samples were removed if they failed quality assurance based on their detection p-values. All samples were Illumina and SWAN normalised to reduce technical bias between type 1 and type 2 probes.

### Confounding variables

Directed acyclic graphs were constructed (online supplemental figures 1 and 2) from which a minimal set of adjusted variables were selected using the R packages ggdag and dagitty. In the primary analyses, the final models were adjusted for child age, alcohol use in pregnancy, tobacco use in pregnancy, ethnic group, parity, age at delivery and maternal education. All methylation analyses were additionally adjusted for white blood cell composition, using the algorithm by Houseman *et al*.[15]

### Statistical analyses

Outcomes were log-transformed for anaylses. Linear mixed models[16] with random intercepts (one for each offspring) were used to analyse the association between these longitudinal outcome variables and various exposure variables (individual adverse events, the number of such events, the perceived impact of such events, and methylation variables). Missing confounders were imputed as the sample mean of the variable. Sub-analyses were also conducted to estimate the association between the exposures and the log-transformed CV measures at each age separately, and linear regression was used for these analyses instead of linear mixed models and adjusted for the minimal set of potential confounders. All estimates of associations for CV measures are for a 4-unit change in maternal adversities, which corresponds to the difference between the adversity not occurring and the adversity having its highest impact. Linear mixed models[16] were used to test the associations between individual CpG sites with maternal adversity measures and child CV measures. The Bonferroni p value threshold was used to correct for multiple testing in the analyses of individual methylation probes.

P values were based on the likelihood ratio statistic except for the descriptive analyses, where p values for a sex difference were based on a t-test (for continuous variables) or Fisher's exact test (for binary variables).[17] All analyses were conducted in R V.4.0.0.[18]

### Patient and public involvement

There was no direct involvement from participants in the study design. Select participants are part of a committee which meets to discuss and provide insights on acceptability, and study methodology and design. This committee was not involved in the formulation of the current research question and analyses.

### RESULTS

Characteristics and summary data of the sample are as shown in table 1. The median number of maternal events

**Table 1** Participant characteristics at each follow-up

| | n | Pooled sample Mean (±SD)/median (IQR)/n (%) | Boys Mean (±SD)/median (IQR)/n (%) | Girls Mean (±SD)/median (IQR)/n (%) | P for sex difference |
|---|---|---|---|---|---|
| **Pregnancy and birth measures (n=14 901)** | | | | | |
| Maternal age (years at birth) | 12 921 | 28 (5) | 28.1 (5) | 27.9 (4.9) | 0.009 |
| Maternal smoking status n (%) yes | 11 052 | 2157 (19.5%) | 1144 (20.2%) | 1013 (18.8%) | 0.08 |
| Gestation length (weeks) | 12 921 | 39.8 (1.3) | 39.7 (1.3) | 39.8 (1.3) | <0.001 |
| No of events | 12 285 | 3.6 (2.3) | 3.6 (2.3) | 3.6 (2.3) | 0.4 |
| Perceived impact score | 12 285 | 8.5 (7) | 8.4 (7) | 8.6 (7.1) | 0.2 |
| Birth weight (g) | 12 766 | 3469 (478) | 3530 (490) | 3404 (457) | <0.001 |
| Breastfed (% yes) | 10 359 | 6185 (59.7%) | 3132 (59%) | 3053 (60.5%) | 0.1 |
| **Offspring 7 years follow-up (n=7431)** | | | | | |
| Systolic BP | 7065 | 98.8 (9.2) | 98.7 (9.1) | 98.9 (9.3) | 0.4 |
| Diastolic BP | 7063 | 56.5 (6.7) | 56.1 (6.7) | 56.9 (6.6) | <0.001 |
| HR | 7062 | 83.3 (10.7) | 82 (10.5) | 84.6 (10.8) | <0.001 |
| **Offspring 17 years follow-up (n=5215)** | | | | | |
| Systolic BP | 4104 | 116.4 (9.9) | 122 (9.2) | 112 (8.1) | <0.001 |
| Diastolic BP | 4104 | 64.2 (6) | 63.3 (6) | 64.9 (5.9) | <0.001 |
| HR | 4104 | 65.2 (9.7) | 62.5 (9.2) | 67.2 (9.6) | <0.001 |
| cIMT | 4102 | 0.48 (0.05) | 0.48 (0.05) | 0.47 (0.04) | <0.001 |
| PWV | 3423 | 5.8 (0.7) | 6 (0.7) | 5.6 (0.6) | <0.001 |

Maternal smoking is yes/no smoked cigarettes regularly in the last 2 months of pregnancy
Breastfeeding is yes/no 1+ months of breastfeeding
BP, blood pressure; cIMT, carotid intermedia thickness; HR, heart rate; PWV, pulse wave velocity.

and perceived impact score was 3.6 (2.3) and 8.5 (7), respectively (table 1). The most common event during the study period was an argument with partner (63.1%), followed by fetal testing (52.6%) and reductions in income (50.6%) (online supplemental figure 4).

### Maternal adversity and overall offspring CV measures

There was no association between number of events and any of the primary offspring CV measures (CV time points combined) (table 2). Results did not differ when analyses were rerun using perceived impact scores.

### Maternal adversity and offspring CV measures by sex and specific time points

In contrast to our hypotheses, there was an association between perceived impact score and PWV in boys, whereby a four-unit increase in adversity score was associated with a 0.1% decrease in PWV (0.999-fold change, 95% CI 0.997 to 1.001; table 2). When HR and BP measures were examined at specific time points (ie, 7 years and 17 years separately) there was an association between maternal number of events and offspring SBP at 7 years of age in girls, whereby there was a 0.6% decrease in BP for each additional four events (0.994-fold change, 95% CI 0.988 to 0.999). In line with our hypotheses, there was also an association between offspring DBP at 17 years of age

and maternal perceived impact score in girls whereby a four-unit increase in impact score was associated with a 0.2% increase in DBP (1.002-fold change 95% CI 1.000 to 1.005). There were no other associations detected with number of events or perceived impact score at specific time points (data not shown).

### DNA methylation and offspring CV measures

In line with our hypotheses, there were some associations evident with specific CpG sites. In the longitudinal analyses, with timepoints combined, methylation of cg20111643 (TOM1L1) was associated with offspring SBP (1.013-fold change 95% CI 1.008 to 1.017 per SD). There was an association with methylation of cg07494499 (NXN) (1.012-fold change 95% CI: 1.008 to 1.017 per SD of the outcome) and cg02458152 (EFCAB1) (1.011-fold change 95% CI 1.007 to 1.015 per SD) and SBP. There was also an association between methylation of cg20222926 (FEZF1) (0.987-fold change 95% CI 0.982 to 0.992 per SD) and DBP that appeared to be largely driven by rare, large DNA methylation changes (figure 2). However, when the three outliers were excluded, the effect was no longer observed. There were no associations with any other CpG site.

**Table 2** Associations between maternal adversity and offspring CV measures

| Outcome | Exposure | Pooled Fold change (95% CI) | Boys Fold change (95% CI) | Girls Fold change (95% CI) |
|---|---|---|---|---|
| Resting heart rate (bpm) | Perceived impact score | 0.999 (0.998 to 1.001) | 1.000 (0.997 to 1.002) | 0.999 (0.997 to 1.001) |
| | Adversity no | 0.997 (0.992 to 1.003) | 0.998 (0.990 to 1.005) | 0.996 (0.989 to 1.004) |
| Systolic blood pressure (mm Hg) | Perceived impact score | 1.000 (0.999 to 1.001) | 1.000 (0.998 to 1.002) | 1.000 (0.998 to 1.001) |
| | Adversity no | 0.998 (0.994 to 1.001) | 0.998 (0.993 to 1.003) | 0.998 (0.993 to 1.002) |
| Diastolic blood pressure (mm Hg) | Perceived impact sficore | 1.000 (0.999 to 1.002) | 0.999 (0.997 to 1.002) | 1.001 (0.999 to 1.003) |
| | Adversity no | 0.999 (0.994 to 1.003) | 0.996 (0.989 to 1.002) | 1.001 (0.996 to 1.007) |
| Pulse-wave velocity | Perceived impact score | 0.999 (0.997 to 1.001) | 0.999 (0.997 to 1.001) | 1.001 (0.998 to 1.004) |
| | Adversity no | 0.998 (0.991 to 1.005) | 0.9928 (0.982 to 1.004) | 1.001 (0.992 to 1.010) |
| Carotid Intima Media Thickness | Perceived impact score | 1.000 (0.998 to 1.002) | 1.001 (0.998 to 1.004) | 1.000 (0.997 to 1.002) |
| | Adversity no | 1.001 (0.996 to 1.007) | 1.001 (0.993 to 1.010) | 1.001 (0.994 to 1.008) |

Models adjusted for child age, alcohol use in pregnancy; tobacco use in pregnancy; ethnic group; parity; age at delivery and maternal education.
Fold changes corresponds to a four unit change in adversity measures.
NB, pulse wave velocity and carotid Intima media thickness were only measured at one time point (17 years of age).
CV, cardiovascular.

## Specific adversities and offspring CV measures

We further explored how events clustered (online supplemental figure 3), and whether specific events were associated with offspring CV measures stratified by sex (table 3).

In contrast to the hypothesis, at the seven-year follow-up, female offspring of mothers who were admitted to hospital had a 2.5% lower HR (0.975 fold-change 95% CI: 0.956 to 0.994), 1.8% lower SBP (0.982 fold-change 95% CI: 0.967 to 0.996) and 2.6% lower DBP (0.974 fold-change 95% CI 0.957 to 0.992); argued with partner had a 2.4% lower HR (0.976 fold-change 95% CI 0.961 to 0.992) and a 1.5% lower DBP (0.985 fold-change 95% CI 0.971 to

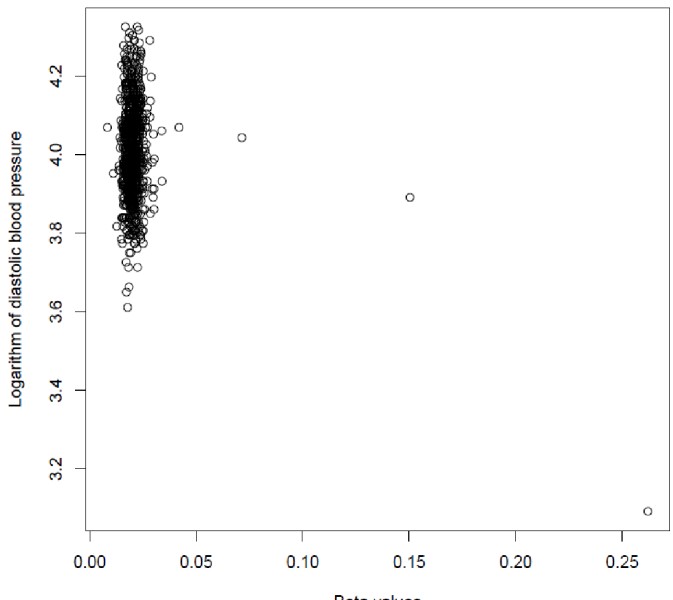

**Figure 2** The relationship between CpG probe cg20222926 and offspring diastolic blood pressure.

0.0.999); argued with family and friends had a 1.8% lower SBP (0.982 fold-change 95% CI 0.965 to 0.999); took an exam had a 4.2% lower SBP (0.958 fold-change 95% CI 0.928 to 0.988); had a partner emotionally cruel to child had a 6.7% lower DBP (0.933 fold-change 95% CI 0.875 to 0.996) and had a partner emotionally cruel to her had a 2.9% lower HR (0.971 fold-change 95% CI 0.944 to 0.999). In agreement with the hypothesis, at seven years female offspring of mothers who were in trouble with the law had a 9.0% higher SBP (1.090 fold-change 95% CI 1.003 to 1.185); were separated had a 3.1% higher SBP (1.031 fold-change 95% CI 1.005 to 1.058); and tried to have an abortion had a 10.4% higher DBP (1.104 fold-change 95% CI 1.012 to 1.205).

In contrast to the hypotheses, at the seven-year follow-up, male offspring of mothers those who had an ill partner had a 2.9% lower SBP (0.971 fold-change 95% CI 0.951 to 0.992); became homeless had a 7.3% lower DBP (0.927 fold-change 95% CI: 0.88 to 0.978); were convicted of an offence had a 14.0% lower SBP (0.860 fold-change 95% CI: 0.743 to 0.996); were in trouble with the law had a 10.4% lower SBP (0.896 fold-change 95% CI 0.816 to 0.983); and had a partner who hurt them had a 6.1% lower DBP (0.939 fold-change 95% CI 0.999 to 0.993). In agreeance, at seven years male offspring who had mothers who attempted suicide had a 25.0% higher HR (1.250 fold-change 95% CI 1.018 to 1.535) and had income reduced had a 1.4% higher SBP (1.014 fold-change 95% CI 1.002 to 1.027).

At 17 years of age, female offspring of mothers who were admitted to hospital had a 2.0% higher DBP (1.020 fold-change 95% CI 1.003 to 1.038); who attempted suicide had a 41.6% higher HR (1.416 fold-change 95% CI 1.081 to 1.854); had moved house had a 4.5% higher HR (1.045 fold-change 95% CI 1.012 to 1.079); whose partner was

**Table 3** Specific maternal adversities and longitudinal offspring CV measures

| Adversity event | Heart rate | | | Systolic blood pressure | | | Diastolic blood pressure | | |
|---|---|---|---|---|---|---|---|---|---|
| | Pooled Fold change (95% CI) | Boys Fold change (95% CI) | Girl Fold change (95% CI) | Pooled Fold change (95% CI) | Boys Fold change (95% CI) | Girls Fold change (95% CI) | Pooled Fold change (95% CI) | Boys Fold change (95% CI) | Girls Fold change (95% CI) |
| Partner died | 0.98 (0.85 to 1.11) | 0.98 (0.82 to 1.17) | 0.98 (0.79 to 1.12) | 0.98 (0.90 to 1.07) | 0.93 (0.83 to 1.05) | 1.04 (0.90 to 1.19) | 0.99 (0.89 to 1.11) | 0.95 (0.82 to 1.10) | 1.06 (0.90 to 1.26) |
| Child died | 1.06 (0.85 to 1.33) | 1.06 (0.70 to 1.60) | 1.06 (0.81 to 1.40) | 1.10 (0.94 to 1.27) | 1.08 (0.82 to 1.41) | 1.11 (0.94 to 1.33) | 1.15 (0.96 to 1.38) | 1.18 (0.84 to 1.67) | 1.14 (0.92 to 1.41) |
| Friend or relative died | 1.00 (0.98 to 1.01) | 1.00 (0.98 to 1.02) | 0.99 (0.97 to 1.01) | 0.99 (0.98 to 1.00) | 1.00 (0.98 to 1.01) | 0.99 (0.98 to 1.00) | 1.00 (0.99 to 1.01) | 1.00 (0.98 to 1.01) | 1.00 (0.99 to 1.02) |
| Child was ill | 1.00 (0.98 to 1.01) | 1.00 (0.98 to 1.02) | 0.99 (0.97 to 1.01) | 1.00 (0.99 to 1.01) | 1.00 (0.99 to 1.02) | 0.99 (0.98 to 1.00) | 1.00 (0.98 to 1.01) | 1.00 (0.98 to 1.02) | 0.99 (0.98 to 1.01) |
| Partner was ill | 1.00 (0.98 to 1.02) | 1.00 (0.98 to 1.03) | 1.00 (0.97 to 1.02) | 0.99 (0.98 to 1.00) | 0.98 (0.97 to 1.00) | 1.00 (0.98 to 1.02) | 1.00 (0.98 to 1.01) | 0.99 (0.96 to 1.01) | 1.01 (0.99 to 1.03) |
| Friend or relative was ill | 1.00 (0.98 to 1.01) | 1.00 (0.98 to 1.02) | 0.99 (0.97 to 1.01) | 1.00 (0.99 to 1.01) | 1.00 (0.99 to 1.01) | 1.00 (0.99 to 1.01) | 1.01 (0.99 to 1.02) | 1.01 (0.99 to 1.02) | 1.00 (0.99 to 1.02) |
| Admitted to hospital | 1.00 (0.98 to 1.01) | 1.01 (0.99 to 1.03) | 0.98 (0.97 to 1.00) | 1.00 (0.99 to 1.00) | 1.00 (0.99 to 1.02) | 0.99 (0.98 to 1.00)* | 1.00 (0.99 to 1.01) | 1.00 (0.99 to 1.02) | 0.99 (0.97 to 1.01) |
| In trouble with the law | 1.03 (0.95 to 1.10) | 1.06 (0.94 to 1.19) | 1.00 (0.94 to 1.19) | 1.00 (0.96 to 1.05) | 0.91 (0.84 to 0.99)* | 1.06 (1.00 to 1.12) | 1.01 (0.95 to 1.07) | 0.97 (0.88 to 1.07) | 1.04 (0.97 to 1.11) |
| Divorced | 1.03 (0.96 to 1.10) | 1.03 (0.95 to 1.12) | 0.99 (0.89 to 1.10) | 1.04 (1.00 to 1.08) | 1.03 (0.97 to 1.09) | 1.06 (1.00 to 1.12) | 1.05 (0.99 to 1.10) | 1.04 (0.97 to 1.11) | 1.05 (0.98 to 1.13) |
| Partner rejected pregnancy | 1.0 (0.98 to 1.03) | 0.98 (0.95 to 1.02) | 1.20 (0.99 to 1.06) | 0.99 (0.98 to 1.01) | 0.98 (0.96 to 1.00) | 1.01 (0.99 to 1.03) | 1.01 (0.99 to 1.03) | 0.99 (0.97 to 1.02) | 1.02 (1.00 to 1.05) |
| Very ill | 1.01 (0.99 to 1.03) | 1.02 (0.99 to 1.04) | 1.00 (0.98 to 1.03) | 1.01 (1.00 to 1.02) | 1.01 (0.99 to 1.03) | 1.01 (0.99 to 1.03) | 1.02 (1.01 to 1.04)† | 1.03 (1.01 to 1.05)† | 1.01 (0.99 to 1.03) |
| Partner lost job | 1.00 (0.98 to 1.01) | 1.00 (0.97 to 1.03) | 0.99 (0.97 to 1.02) | 1.00 (0.99 to 1.01) | 1.01 (0.99 to 1.02) | 1.00 (0.98 to 1.02) | 1.01 (1.00 to 1.03) | 1.01 (0.99 to 1.04) | 1.01 (0.99 to 1.03) |
| Partner had problems at work | 0.99 (0.98 to 1.01) | 0.99 (0.97 to 1.01) | 1.00 (0.98 to 1.01) | 1.00 (0.99 to 1.01) | 1.00 (0.99 to 1.01) | 1.00 (0.99 to 1.01) | 1.00 (0.99 to 1.01) | 0.99 (0.97 to 1.00) | 1.00 (0.99 to 1.01) |
| Problems at work | 1.00 (0.98 to 1.01) | 0.99 (0.98 to 1.02) | 1.00 (0.97 to 1.02) | 1.01 (1.00 to 1.02) | 1.01 (0.99 to 1.03) | 1.01 (0.99 to 1.02) | 1.01 (1.00 to 1.02) | 1.01 (0.99 to 1.03) | 1.02 (1.00 to 1.03) |
| Lost job | 0.99 (0.96 to 1.02) | 0.97 (0.93 to 1.02) | 1.00 (0.96 to 1.04) | 1.00 (0.98 to 1.02) | 1.01 (0087 to 1.04) | 1.00 (0.98 to 1.03) | 1.01 (0.99 to 1.03) | 1.01 (0.98 to 1.05) | 1.01 (0.98 to 1.04) |
| Partner went away | 1.00 (0.98 to 1.02) | 1.00 (0.97 to 1.02) | 1.00 (0.98 to 1.02) | 1.00 (0.99 to 1.01) | 0.99 (0.98 to 1.01) | 1.00 (0.99 to 1.02) | 1.00 (0.98 to 1.01) | 0.99 (0.97 to 1.01) | 1.00 (0.99 to 1.02) |
| Partner in trouble with law | 0.98 (0.95 to 1.02) | 1.00 (0.95 to 1.06) | 0.97 (0.92 to 1.02) | 1.00 (0.98 to 1.03) | 0.99 (0.95 to 1.03) | 1.02 (0.99 to 1.06) | 0.98 (0.95 to 1.01) | 0.95 (0.91 to 1.00)* | 1.00 (0.96 to 1.04) |
| Separated | 1.01 (0.99 to 1.03) | 1.01 (0.98 to 1.04) | 1.01 (0.98 to 1.04) | 1.00 (0.99 to 1.02) | 0.99 (0.97 to 1.01) | 1.03 (1.01 to 1.04)† | 1.00 (0.98 to 1.02) | 0.99 (0.96 to 1.02) | 1.01 (0.99 to 1.04) |
| Income reduced | 1.00 (0.99 to 1.01) | 1.01 (0.99 to 1.02) | 0.99 (0.97 to 1.00) | 1.00 (1.00 to 1.01) | 1.01 (1.00 to 1.02)* | 1.00 (0.99 to 1.01) | 1.01 (1.00 to 1.02)* | 1.01 (1.00 to 1.03) | 1.01 (0.99 to 1.02) |

Continued

**Table 3** Continued

| Adversity event | Heart rate | | | Systolic blood pressure | | | Diastolic blood pressure | | |
|---|---|---|---|---|---|---|---|---|---|
| | Pooled Fold change (95% CI) | Boys Fold change (95% CI) | Girl Fold change (95% CI) | Pooled Fold change (95% CI) | Boys Fold change (95% CI) | Girls Fold change (95% CI) | Pooled Fold change (95% CI) | Boys Fold change (95% CI) | Girls Fold change (95% CI) |
| Argued with partner | 0.99 (0.98 to 1.00) | 1.00 (0.99 to 1.02) | 0.98 (0.97 to 1.00)* | 1.00 (0.99 to 1.00) | 1.00 (0.99 to 1.01) | 0.99 (0.98 to 1.00) | 1.00 (0.99 to 1.01) | 1.00 (0.99 to 1.01) | 0.99 (0.98 to 1.00) |
| Argued with family or friends | 1.00 (0.98 to 1.01) | 1.00 (0.97 to 1.02) | 1.00 (0.98 to 1.02) | 1.00 (0.99 to 1.01) | 1.00 (0.99 to 1.02) | 0.99 (0.98 to 1.00) | 1.00 (0.99 to 1.01) | 1.00 (0.98 to 1.01) | 1.00 (0.99 to 1.02) |
| Moved house | 1.02 (1.00 to 1.03)* | 1.01 (0.99 to 1.04) | 1.02 (1.00 to 1.04)* | 1.00 (0.99 to 1.01) | 1.00 (0.99 to 1.02) | 1.00 (0.99 to 1.02) | 0.99 (0.98 to 1.01) | 0.99 (0.97 to 1.01) | 1.00 (0.98 to 1.01) |
| Partner hurt mother | 0.99 (0.95 to 1.03) | 0.99 (0.94 to 1.05) | 0.99 (0.94 to 1.04) | 0.99 (0.97 to 1.02) | 0.98 (0.95 to 1.02) | 1.01 (0.98 to 1.04) | 0.97 (0.94 to 1.00)* | 0.95 (0.90 to 0.99)* | 0.98 (0.94 to 1.02) |
| Became homeless | 1.00 (0.97 to 1.04) | 1.00 (0.95 to 1.06) | 1.00 (0.95 to 1.05) | 1.00 (0.98 to 1.03) | 0.99 (0.96 to 1.03) | 1.01 (0.98 to 1.05) | 0.97 (0.94 to 1.00)* | 0.94 (0.90 to 0.98)† | 1.00 (0.96 to 1.04) |
| Major financial problems | 1.01 (1.00 to 1.02) | 1.01 (0.99 to 1.03) | 1.01 (0.99 to 1.02) | 1.01 (1.00 to 1.02) | 1.01 (1.00 to 1.02) | 1.01 (1.00 to 1.02) | 1.01 (1.00 to 1.02) | 1.01 (0.99 to 1.02) | 1.01 (1.00 to 1.03)* |
| Got married | 1.00 (0.97 to 1.03) | 0.98 (0.94 to 1.03) | 1.02 (0.97 to 1.07) | 1.00 (0.97 to 1.01) | 1.00 (0.97 to 1.02) | 0.99 (0.96 to 1.02) | 1.00 (0.98 to 1.03) | 1.00 (0.96 to 1.03) | 1.01 (0.97 to 1.05) |
| Partner hurt child | 0.94 (0.82 to 1.08) | 0.73 (0.59 to 0.91)* | 0.73 (0.59 to 0.91)† | 1.03 (0.94 to 1.11) | 0.93 (0.81 to 1.07) | 1.06 (0.96 to 1.18) | 0.95 (0.85 to 1.05) | 0.87 (0.73 to 1.04) | 1.00 (0.88 to 1.14) |
| Attempted suicide | 1.20 (1.05 to 1.39)† | 1.26 (1.02 to 1.56)* | 1.16 (0.95 to 1.40) | 0.97 (0.88 to 1.06) | 1.01 (0.88 to 1.16) | 0.96 (0.85 to 1.08) | 1.07 (0.96 to 1.21) | 1.09 (0.92 to 1.31) | 1.06 (0.91 to 1.23) |
| Convicted of an offence | 1.09 (0.97 to 1.23) | 1.14 (0.93 to 1.39) | 1.07 (0.93 to 1.23) | 0.99 (0.92 to 1.07) | 0.86 (0.75 to 0.98)* | 1.07 (0.97 to 1.17) | 1.02 (0.93 to 1.12) | 0.98 (0.83 to 1.15) | 1.04 (0.94 to 1.17) |
| Bled & thought might miscarry | 0.99 (0.97 to 1.00) | 0.99 (0.97 to 1.01) | 0.99 (0.97 to 1.01) | 1.00 (1.00 to 1.01) | 1.00 (0.99 to 1.02) | 1.00 (0.99 to 1.02) | 1.00 (0.99 to 1.01) | 1.00 (0.98 to 1.01) | 1.00 (0.99 to 1.02) |
| Started new job | 1.00 (0.97 to 1.04) | 0.99 (0.94 to 1.04) | 1.01 (0.97 to 1.06) | 0.99 (0.97 to 1.02) | 0.99 (0.95 to 1.02) | 0.99 (0.96 to 1.02) | 1.00 (0.99 to 1.01) | 0.99 (0.97 to 1.00) | 1.00 (0.99 to 1.01) |
| Test to see if baby abnormal | 1.00 (0.98 to 1.01) | 0.99 (0.97 to 1.01) | 1.00 (0.98 to 1.02) | 1.00 (0.99 to 1.01) | 0.99 (0.98 to 1.00) | 1.00 (0.99 to 1.01) | 1.00 (0.99 to 1.01) | 0.99 (0.97 to 1.00) | 1.00 (0.99 to 1.02) |
| Tests show baby possibly abnormal | 1.00 (0.97 to 1.02) | 0.99 (0.96 to 1.02) | 1.01 (0.98 to 1.04) | 1.00 (0.98 to 1.01) | 0.99 (0.96 to 1.01) | 1.01 (0.99 to 1.03) | 1.00 (0.98 to 1.02) | 0.99 (0.96 to 1.01) | 1.01 (0.98 to 1.04) |
| Told having twins | 1.05 (1.00 to 1.11) | 1.03 (0.95 to 1.10) | 1.09 (1.00 to 1.19)* | 0.99 (0.96 to 1.03) | 0.99 (0.95 to 1.04) | 0.99 (0.94 to 1.04) | 0.98 (0.94 to 1.02) | 0.97 (0.92 to 1.03) | 0.99 (0.92 to 1.06) |
| Possible harm to baby | 0.99 (0.97 to 1.01) | 0.98 (0.95 to 1.00) | 1.00 (0.97 to 1.03) | 1.00 (0.99 to 1.02) | 1.00 (0.98 to 1.01) | 1.01 (0.99 to 1.03) | 1.00 (0.99 to 1.02) | 0.99 (0.97 to 1.02) | 1.02 (0.99 to 1.04) |
| Tried to have abortion | 1.04 (0.98 to 1.10) | 1.04 (0.96 to 1.14) | 1.03 (0.95 to 1.12) | 1.01 (0.98 to 1.06) | 0.99 (0.94 to 1.05) | 1.04 (0.98 to 1.09) | 1.02 (0.98 to 1.08) | 1.03 (0.96 to 1.11) | 1.03 (0.96 to 1.10) |
| Took an exam | 1.01 (0.99 to 1.04) | 1.05 (1.00 to 1.09)* | 0.99 (0.95 to 1.03) | 0.99 (0.97 to 1.01) | 1.01 (0.98 to 1.04) | 0.97 (0.94 to 0.99)† | 0.99 (0.97 to 1.01) | 1.01 (0.97 to 1.04) | 0.98 (0.94 to 1.01) |
| Partner emotionally cruel to mother | 0.99 (0.97 to 1.01) | 0.98 (0.96 to 1.00) | 1.00 (0.97 to 1.02) | 1.00 (0.99 to 1.02) | 1.01 (0.99 to 1.02) | 1.00 (0.98 to 1.02) | 1.01 (0.99 to 1.02) | 1.00 (0.99 to 1.02) | 1.01 (0.99 to 1.03) |

Continued

**Table 3** Continued

| Adversity event | Heart rate | | | Systolic blood pressure | | | Diastolic blood pressure | | |
|---|---|---|---|---|---|---|---|---|---|
| | Pooled Fold change (95% CI) | Boys Fold change (95% CI) | Girl Fold change (95% CI) | Pooled Fold change (95% CI) | Boys Fold change (95% CI) | Girls Fold change (95% CI) | Pooled Fold change (95% CI) | Boys Fold change (95% CI) | Girls Fold change (95% CI) |
| Partner emotionally cruel to child | 0.99 (0.95 to 1.04) | 1.01 (0.93 to 1.09) | 0.98 (0.92 to 1.05) | 0.99 (0.96 to 1.02) | 0.98 (0.93 to 1.03) | 0.99 (0.96 to 1.04) | 0.97 (0.94 to 1.01) | 1.01 (0.94 to 1.08) | 0.96 (0.91 to 1.00) |
| House or car burgled | 1.00 (0.97 to 1.02) | 0.98 (0.95 to 1.02) | 1.00 (0.97 to 1.03) | 0.99 (0.97 to 1.00) | 1.00 (0.97 to 1.02) | 0.99 (0.97 to 1.01) | 0.99 (0.98 to 1.01) | 0.98 (0.96 to 1.01) | 1.00 (0.97 to 1.02) |
| Had an accident | 1.01 (0.98 to 1.05) | 1.00 (0.95 to 1.04) | 1.03 (0.98 to 1.08) | 0.99 (0.97 to 1.01) | 0.98 (0.96 to 1.01) | 0.99 (0.96 to 1.02) | 1.00 (0.97 to 1.02) | 0.99 (0.96 to 1.03) | 1.00 (0.96 to 1.03) |

NB, Fold changes represent a four unit change.

*Denotes p≤0.05
†Denotes p≤0.01
‡Denotes p≤0.001

emotional cruel to them had a 2.1% (1.021 fold-change 95% CI 1.002 to 1.041) and 3.1% (1.031 fold-change 95% CI 1.006 to 1.056) higher SBP and DBP, respectively; whose partner rejected the pregnancy had a 7.2% higher HR (1.072 fold-change 95% CI 1.019 to 1.127) and was very ill has a 2.5% higher SBP (1.025 fold-change 95% CI 1.004 to 1.046).

At 17 years of age, male offspring who had mothers that were admitted to hospital had a 2.2% higher DBP (1.022 fold-change 95% CI 1.001 to 1.043); were convicted of an offence had a 91.9% higher HR (1.919 fold-change 95% CI 1.209 to 3.045); had major financial problems had a 2.8% higher DBP (1.028 fold-change 95% CI 1.005 to 1.051); was told they were having twins had a 7.2% higher HR (1.072 fold-change 95% CI 1.006 to 1.144); took an exam had a 9.8% higher HR (1.098 fold-change 95% CI 1.027 to 1.174) and was very ill had a 3.2% (1.032 fold-change 95% CI 1.009 to 1.056) and 4.2% (1.042 fold-change 95% CI 1.013 to 1.072) higher SBP and DBP, respectively. Contrary to the hypothesis at 17 years of age, male offspring of mothers who argued with partner, had a partner who was emotionally cruel to them, had a partner who hurt their child, had a partner reject the pregnancy or possible harm to the baby had a 2.7% (0.973 fold-change 95% CI 0.948 to 0.999), 4.5% (0.955 fold-change 95% CI 0.915 to 0.998), 34.5% (0.655 fold-change 95% CI 0.492 to 0.873), 7.7% (0.923 fold-change 95% CI 0.0.865 to 0.984) and 5.0% (0.950 fold-change 95% CI 0.902 to 0.996) lower HR, respectively.

## DISCUSSION

There was no evidence of an overall association between our primary CV measures in offspring and maternal adversity. There was limited evidence to suggest that subtypes of adversity or specific may be associated with CV measures in an age-specific manner as well as an association between CV measures and some CpG probes.

Associations between adversity and health outcomes previously reported in the literature are thought to be moderated by biological changes induced by the stress response. Global methylation is associated with CVD in adult populations.[19] However, the association between epigenetic changes at birth and CV measures in childhood and adolescence is less well characterised. It could be that infancy and childhood is a more sensitive period to CV changes induced by adversity than during pregnancy and, although previous results are mixed. For instance, there is support for associations between childhood maltreatment and CVD and risk factors in adulthood.[20] It is possible that exposure to adversities experienced by this cohort were not severe, nor prolonged enough, to have a direct impact on DNA methylation and/or on cardiac function. There were some associations with specific CpG sites, cg20111643 (TOM1L1), cg07494499 (NXN), cg02458152 (EFCAB1) and cg20222926 (FEZF1). Of the genes that these sites are located on only one, NXN, has a postulated role in cardiac development through its role in

the canonical Wnt/β-catenin signalling pathway.[21] Interestingly EFCAB1 has also been implicated in BP measurements,[22] as was observed in this cohort. Of note is the association with cg20222926 (FEZF1), which may be the result of interesting biology, or could be a consequence of measurement error. Future investigations should also consider whether factors such as exposure to maternal hypertensive disorders in utero, such as pre-eclampsia, may play a role in the causal pathway of any observed associations.

Few studies have looked at maternal adversity and CV risk factors in childhood and adolescence. Within this cohort, no association was observed between childhood adversity and BP at 7 and 11 years of age.[23] In an Australian cohort of children those with lower psychosocial stress had higher pulse pressure at age 11.[24] This finding is similar to the favourable associations observed in our study between specific adversities and offspring CV measures, at 7 years of age. Given that this is a paediatric study population it is possible that the unexpected increases in BP observed at 7 years of age may be a feature of the developing CV system in the offspring.[25] Of further consideration is that CV measures during childhood and adolescence may not wholly predict progression to CVD in adulthood.[26] Thus, the results presented do not preclude further examination of perinatal adversity and CVD and risk in adulthood. However, while these measures do not wholly predict progression during adulthood the observed associations between maternal adversity and offspring CV markers, such as BP and PWV, may be early evidence of CV dysfunction. It is plausible that the risk pathways between maternal stress and CVD risk are activated, but the full extent of damage is not yet evident. This would be consistent with the accumulation hypothesis of lifecourse epidemiology, which purports that health disparities become more pronounced with age (ie, diverge).[27] Moreover, the measure of maternal adversity used in this study was an inventory of life events, not based on a conceptual framework, such as that of the adverse childhood experiences construct. Thus, this measure of adversity may not have captured all stressors during pregnancy, which may conceal a legitimate association and in part explain the null findings. Lastly, emerging evidence has suggested that the human placenta may buffer the effects of maternal stress and protect the developing fetus,[5] which could provide a biological explanation for apparent absent effects of maternal stress in this cohort.

Specific adversities were largely associated with favourable changes in offspring CV measures at age 7. At age 17, the direction of the association largely reversed, most pronounced in females. This is suggestive of a protective adaptive response to maternal adversity present in childhood that may reverse trajectory by age 17. Contrary to the original hypotheses, at age 7, specific maternal adversities largely appeared to have a protective effect on offspring CV measures. Similarly, in this same cohort, a different study observed that maternal prenatal anxiety and depressive symptomology was inversely associated with offspring BP at 10–11 years of age, although to a similar magnitude as paternal measures.[28] However, given this association was not looked at beyond 11 years of age it is not known if a similar reversal of trajectory was present at 17 years. Given multiple comparisons, it is also possible that the associations between specific adversities and offspring CV have arisen due to chance. However, it is curious that the associations largely follow the same age-trajectory, that is an inverse association with adversity events at 7 years and a positive association at 17 years. It is also noteworthy that reported adversities that had the largest effect size were those that would presumably have more psychological impact for example, partner hurt child and mother convicted of an offence. Nevertheless, replication in other cohorts would have to be demonstrated to confirm such associations.

A strength of the current study is its large sample size and its detailed collection of longitudinal phenotypic data in both mothers and their children followed into adolescence. However, as is the case with such long-term observational studies, over time, there is evidence of attrition, which may introduce bias, with those who were of a higher socioeconomic position being more likely to remain in the study over time thus potentially limiting the generalisabilty of the results. Moreover, the list of potentially life stressors was not exhaustive and may have resulted in measurement error influencing the results. In addition, the adversity scores calculated as part of this study have not been previously validated. Furthermore, to capture maternal adversity during pregnancy, we took the weighted average of the Life Events Inventory, which was inclusive from the beginning of pregnancy to 8 weeks post-partum). Thus, any effects may have been diluted by the inclusion of adversity in the 8 weeks post birth during the perinatal period. Lastly, future studies may benefit from the examination of specific key genes that have been identified in CVD pathways aside from global methylation measures.

In summary, the results presented largely do not support an association between maternal prenatal adversity, and offspring methylation and CV measures during childhood and adolescence. There were, however some sex-specific and age-specific trends which would have to be confirmed in future studies. Identification and confirmation of these associations between maternal adversity and offspring CV function may assist with identifying high-risk populations for which additional monitoring may be appropriate.

**Author affiliations**

[1]Deakin University, IMPACT - the Institute for Mental and Physical Health and Clinical Translation,School of Medicine, Barwon Health, Geelong, VIC, Australia

[2]Centre for Epidemiology and Biostatistics, Melbourne School of Population and Global Health, The University of Melbourne, Parkville, Victoria, Australia

[3]Centre for Health Equity, Melbourne School of Population and Global Health, University of Melbourne, Parkville, Victoria, Australia

[4]Centre for Mental Health, Melbourne School of Population and Global Health, University of Melbourne, Melbourne, Victoria, Australia

[5]School of Psychology, Deakin, Geelong, Victoria, Australia
[6]Community Child Health, Murdoch Childrens Research Institute, Parkville, Victoria, Australia

**Contributors** AO'N conceived the initial idea for examining the associations and all authors made a substantial contribution to the conception and/or design of the study analyses. JGD performed all statistical analyses. NKH wrote the initial draft of the manuscript and all authors reviewed (NKH, JGD, AS, GA, GS, LO, KL and AO'N) and contributed intellectual content. All authors have approved of the final version that has been submitted. A'ON acts as guarantor for overall content.

**Funding** The UK Medical Research Council and Wellcome (Grant ref: 217065/Z/19/Z) and the University of Bristol provide core support for ALSPAC. A comprehensive list of grants funding is available on the ALSPAC website http://www.bristol.ac.uk/alspac/external/documents/grant-acknowledgements. pdf This research was specifically funded by the British Heart Foundation who provided support for the collection of pulse wave velocity measures (RG/10/004/28240GWAS). Data was generated by Sample Logistics and Genotyping Facilities at Wellcome Sanger Institute and LabCorp (Laboratory Corporation of America) using support from 23andMe. This specific analyses were supported by a grant from the University of Melbourne.

**Competing interests** None declared.

**Patient consent for publication** Not applicable.

**Ethics approval** Ethical approval for the study was obtained from the ALSPAC Ethics and Law Committee and the Local Research Ethics Committees and The University of Melbourne Human Research Ethics committee (ref: 1853268.1). Consent for biological samples has been collected in accordance with the Human Tissue Act (2004) and informed consent for the use of data collected via questionnaires and clinics was obtained from participants following the recommendations of the ALSPAC Ethics and Law Committee at the time. Participants gave informed consent to participate in the study before taking part.

**Provenance and peer review** Not commissioned; externally peer reviewed.

**Data availability statement** Data are available on reasonable request. The data underlying this article will be shared on reasonable request to the corresponding author with permission from the ALSPAC team in accordance with data sharing agreements. The data underlying this article will be shared on reasonable request to the corresponding author with permission from the ALSPAC team in accordance with data sharing agreements.

**ORCID iDs**
Natalie K Hyde http://orcid.org/0000-0002-0693-2904
Gregory Armstrong http://orcid.org/0000-0002-8073-9213

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
