## [Reviewer comments · BMJ Open]

ARTICLE DETAILS

TITLE (PROVISIONAL)	The association between maternal adversity, DNA methylation, and cardiovascular health of offspring: a longitudinal analysis of the ALSPAC cohort study
AUTHORS	Hyde, Natalie; Dowty, James; Scovelle, Anna; Armstrong, Gregory; Sutherland, Georgina; Olive, Lisa; Lycett, kate; O'Neil, Adrienne

VERSION 1 – REVIEW

REVIEWER	Aditi Bhargava UCSF, CRS and ObGyn
REVIEW RETURNED	29-Jul-2021

GENERAL COMMENTS	In this longitudinal observational cohort study, the authors find that maternal adversities (myriad of them) do not correlate with primary cardiovascular outcomes such as blood pressure. A small correlation was observed with global methylation status of cord blood DNA with systolic blood pressure. It is an important and interesting study. Some concerns noted are below. Perhaps if alcohol consumption and smoking/tobacco use were used as contributing factors rather than adjusting for them, some outcomes may become more significant. This is because stress can drive people to smoke or drink. A. Abstract: A1. Objectives Lines 8-10: Perhaps re-structure to say- Maternal.....associated with some health outcomes in offspring? A2. Results: Sex effects is offspring sex and can be better stated. I commend the authors for not interchangeably using the term sex and gender. Please define SBP or any abbreviation when they first appear. A3. Conclusions: Sex and age of offspring or sex of offspring and maternal age? Please clarify. B. Introduction: B1. Lines 24-27: surely stressors, diet, and environmental factors during one's entire life influence epigenetics as well as interact with one's genome, and thus would influence not just CVD outcomes, but all health and disease outcomes. Please consider re-writing. B2. Lines 47-50: this is a very broad statement - an opinion that is not supported by evidence. Please see a recent publication that provides strong evidence that placenta protects the fetus from
---

	maternal stressors: “Human Placenta Buffers the Fetus from Adverse Effects of Perceived Maternal Stress”, PMID: 33673157. Please modify introduction and discussion accordingly. B3. What would global methylation status really inform us? Shouldn't the authors have focused on methylation status of some key genes in the CVD pathway? C. Methods. C1. Line 15: Please use the term pregnant women and not female. Since this is a human study, use of terminology, such as women or patients is appropriate. C2. Lines 36-40: What is the rationale for adjusting for various factors? Wouldn't maternal age, alcohol or tobacco use actually influence methylation? Wouldn't they be contributing variables rather than confounding? C3: was methylation status also adjusted for total DNA content? D. Results. Overall, too many abbreviations are used and it is harder for a reader not the field to remember so many abbreviations. It would be helpful to provide a rationale statement at the beginning of each result section. D1. What is the significance of association of PWV scores in boys and DBP in girls with maternal adversity? What is the significance of the associations between maternal adversities and CVD parameters? Can the authors better discuss the significance of their findings? D2. What is cg20111643? Is it a gene locus? What is its significance to CVD? Similarly, please provide some context and background on all cg mentioned in this study. These are mentioned in the discussion, but providing some context in the introduction or at the beginning of the results para might be helpful. Otherwise, the reader is clueless about these gene loci until discussion.
--	--

REVIEWER	Xiaotian Li Obstetrics and Gynecology Hospital of Fudan University
REVIEW RETURNED	10-Oct-2021

GENERAL COMMENTS	This is interesting research, which estimated the association between maternal adversity score/cord blood methylation and off-spring (7 and 17 years old) CV health. Generally, it is complex research. Finally, they did not find overall association. There are some concerns about data analysis.  1. could you show us a detailed flowchart about how the studied population got? 2. How did you did to handle the missing data? 3. How many pairs miss to follow-up and how did you handle these data? 4. which cord blood, artery or vein? 5. could you provide pregnancy outcomes, which may relate to changes of methylation of cord blood or off-spring CV.
--

VERSION 1 – AUTHOR RESPONSE

Reviewer: 1

Dr. Aditi Bhargava, UCSF

Comments to the Author:

In this longitudinal observational cohort study, the authors find that maternal adversities (myriad of them) do not correlate with primary cardiovascular outcomes such as blood pressure. A small correlation was observed with global methylation status of cord blood DNA with systolic blood pressure.

It is an important and interesting study. Some concerns noted are below. Perhaps if alcohol consumption and smoking/tobacco use were used as contributing factors rather than adjusting for them, some outcomes may become more significant. This is because stress can drive people to smoke or drink.

A. Abstract:

A1. Objectives Lines 8-10: Perhaps re-structure to say- Maternal.....associated with some health outcomes in offspring?

This has now been restructured to read as follows:

“Maternal adversity during pregnancy has been shown to be associated with some health outcomes in the offspring.”

A2. Results: Sex effects is offspring sex and can be better stated. I commend the authors for not interchangeably using the term sex and gender.

This has now been updated in the Results and Conclusions section of the abstract accordingly as follows:

“Some small offspring sex effects were observed and there was also a small association between methylation of cg20111643 in cord blood and offspring SBP (1.013-fold change 95% CI: 1.008, 1.017 per standard deviation).”

and

“Offspring sex- and age-specific associations require further investigation.”

Please define SBP or any abbreviation when they first appear.

SBP has now been defined where first mentioned in the abstract. To the best of the authors knowledge all other abbreviations have been defined when first referred to in the body of the text.

A3. Conclusions: Sex and age of offspring or sex of offspring and maternal age? Please clarify.

Please refer to response A2.

B. Introduction:

B1. Lines 24-27: surely stressors, diet, and environmental factors during one’s entire life influence epigenetics as well as interact with one’s genome, and thus would influence not just CVD outcomes, but all health and disease outcomes. Please consider re-writing.

The authors acknowledge that these factors would likely affect other health outcomes as well as CVD outcomes. There was only mention to influences on CVD given the focus of the current paper, and given that the seminal work in the DOHaD field was conducted examining CVD outcomes. This has now been rewritten to acknowledge the influence of health outcomes beyond CVD as follows:

“Further extending this work, the Developmental Origins of Health and Disease (DOHaD) hypothesis proposes that the risk of chronic diseases originate not only from an individual’s genome but also by its interactions with biological insults in utero and early life.”

B2. Lines 47-50: this is a very broad statement - an opinion that is not supported by evidence. Please see a recent publication that provides strong evidence that placenta protects the fetus from maternal stressors: “Human Placenta Buffers the Fetus from Adverse Effects of Perceived Maternal Stress”, PMID: 33673157. Please modify introduction and discussion accordingly.

The authors have reviewed the publication referred to and as such have updated the Introduction and Discussion to include this recent evidence as follows:

“However, recent evidence also suggests that perhaps the placenta may buffer the effects of the maternal stress response.”

and

“Lastly, emerging evidence has suggested that the human placenta may buffer the effects of maternal stress and protect the developing fetus, which could provide a biological explanation for apparent absent effects of maternal stress in this cohort.”

This publication is now also cited and included in the reference list accordingly.

B3. What would global methylation status really inform us?

Shouldn’t the authors have focused on methylation status of some key genes in the CVD pathway?

Altered DNA methylation has been associated with CVD risk (<https://journals.plos.org/plosone/article?id=10.1371/journal.pone.0009692>) and in the brain and immune system - the latter when associated with exposure to early life trauma <https://www.sciencedirect.com/science/article/pii/S2352289520300394>. We thus thought it was appropriate but acknowledge in the limitations section that key genes in the CVD pathway may have utility in subsequent studies as follows:

“Lastly, future studies may benefit from the examination of specific key genes that have been identified in CVD pathways aside from global methylation measures.”

C. Methods.

C1. Line 15: Please use the term pregnant women and not female. Since this is a human study, use of terminology, such as women or patients is appropriate.

This line has now been updated to read:

“Briefly, all pregnant women residing in county Avon (~0.9 million) were eligible to participate if their estimated delivery date was between 1 April 1991 and 31 December 1992 inclusive.”

C2. Lines 36-40: What is the rationale for adjusting for various factors? Wouldn’t maternal age,

alcohol or tobacco use actually influence methylation? Wouldn't they be contributing variables rather than confounding?

As referred to in the Methods section, "Directed acyclic graphs (DAGs) were constructed (Supplementary Figures S1 and S2) from which a minimal set of adjusted variables were selected using the R packages *ggdag* and *dagitty*." DAGs are a sophisticated way to select potential confounders while avoiding collider bias, based on a plausible set of causal relationships between variables. However, the true causal relationships are unknown, so any choice of DAG can always be questioned. We note that, at worst, if the variables that you mention are on the causal pathway from maternal adversity to methylation, then our results will be attenuated towards the null (so our results would be conservative) but we believe our current conclusion (that we didn't observe evidence of an association) is still valid, since we do not claim there is no association, rather that we didn't observe any evidence of one.

C3: was methylation status also adjusted for total DNA content?

The main methylation variables were beta values for different methylation sites. Each beta value is the *proportion* of DNA molecules that are methylated at the given methylation site (out of all DNA molecules containing the given methylation site). Beta values are therefore automatically adjusted for the amount of DNA, in the sense that doubling or tripling the amount of DNA leaves the beta value unchanged. So following standard practice, we did not adjust for (i.e. include as a potential confounder in the regression analyses) the amount of DNA in the methylation analyses.

D. Results. Overall, too many abbreviations are used and it is harder for a reader not the field to remember so many abbreviations. It would be helpful to provide a rationale statement at the beginning of each result section.

There are 6 abbreviations used throughout the text of the discussion consistently (CV: cardiovascular, PWV: Pulse wave velocity, SBP: Systolic Blood Pressure, DBP: Diastolic Blood Pressure, HR: Heart rate, CI: Confidence Intervals). Except for PWV, these are easily recognisable acronyms in the cardiovascular/medical field thus it is the authors belief that the intended audience should not have difficulty in recognising these. However, the authors are happy to leave it to the editor's discretion as to whether these abbreviations should be written in full throughout the results section to improve readability.

D1. What is the significance of association of PWV scores in boys and DBP in girls with maternal adversity? What is the significance of the associations between maternal adversities and CVD parameters? Can the authors better discuss the significance of their findings?

Identification and confirmation of these associations between maternal adversity and offspring cardiovascular function may assist with identifying high risk populations for which additional monitoring may be appropriate. This has now been added in the concluding statement.

Furthermore we have expanded upon the discussion to reference the relevance to future disease progression as follows:

"However, while these measures do not wholly predict progression during adulthood the observed associations between maternal adversity and offspring CV markers, such as BP and PWV, may be early evidence of cardiovascular dysfunction. It is plausible that the risk pathways between maternal stress and CVD risk are activated, but the full extent of damage is not yet evident."

D2. What is cg20111643? Is it a gene locus? What is its significance to CVD? Similarly, please

provide some context and background on all cg mentioned in this study. These are mentioned in the discussion, but providing some context in the introduction or at the beginning of the results para might be helpful. Otherwise, the reader is clueless about these gene loci until discussion.

These are specific CpG sites. We acknowledge that this may be confusing for the reader so have now put the associated gene in parentheses directly following their mention. We had not provided context for these CpG sites in the sections preceding the discussion as there were the number of CpG sites that were examined are impractical to list individually with their candidate genes and we did not want to make these a focus of the introduction post hoc.

Reviewer: 2

Prof. Xiaotian Li, Obstetrics and Gynecology Hospital of Fudan University

Comments to the Author:

This is interesting research, which estimated the association between maternal adversity score/cord blood methylation and off-spring (7 and 17 years old) CV health. Generally, it is complex research. Finally, they did not find overall association. There are some concerns about data analysis.

1. could you show us a detailed flowchart about how the studied population got?

We have now included a flowchart of the study population and number of pairs missing specific data at each timepoint (Figure 1).

2. How did you did to handle the missing data?

Mother-child pairs with missing the relevant exposure or outcome were excluded. Missing confounders were imputed as the sample mean of the variable. This has now been noted in the Methods section.

3. How many pairs miss to follow-up and how did you handle these data?

There were up to 10,849 pairs (inclusive of the additional sample that was recruited at 7 years) that did not have complete information, as above in response 2 these pairs were excluded from analyses. We have also made mention of the possibility of this introducing bias due to the attrition in the limitations in the Discussion.

4. which cord blood, artery or vein?

The sample taken was venous cord blood, this has now been noted in the Methods section as follows:

“Venous cord blood at birth was used to assess epigenome-wide methylation levels using the Illumina Infinium® HumanMethylation450 (HM450) BeadChip.”

5. could you provide pregnancy outcomes, which may relate to changes of methylation of cord blood or off-spring CV.

Pregnancy outcomes such as mode of birth have been shown to be associated with changes in methylation in cord blood (<https://bmcpregnancychildbirth.biomedcentral.com/articles/10.1186/s12884-021-03748-y>), but appear not be to be directly associated with offspring CV health. Maternal hypertensive disorders

such as pre-eclampsia may be associated with changes in both. We have now referenced a potential causal role for pre-eclampsia in the Discussion.

“Future investigations should also consider whether factors such as exposure to maternal hypertensive disorders in utero, such as pre-eclampsia, may play a role in the causal pathway of any observed associations.”

VERSION 2 – REVIEW

REVIEWER	Aditi Bhargava UCSF, CRS and ObGyn
REVIEW RETURNED	09-Feb-2022
GENERAL COMMENTS	The authors have addressed my major concerns. I have no further comments.